# Universal Adhesives and Adhesion Modes in Non-Carious Cervical Restorations: 2-Year Randomised Clinical Trial

**DOI:** 10.3390/polym14010033

**Published:** 2021-12-22

**Authors:** Patricia Manarte-Monteiro, Joana Domingues, Liliana Teixeira, Sandra Gavinha, Maria Conceição Manso

**Affiliations:** 1FP-I3ID, Faculty of Health Sciences, University Fernando Pessoa, 4200-150 Porto, Portugal; joanad@ufp.edu.pt (J.D.); lilianat@ufp.edu.pt (L.T.); sgavinha@ufp.edu.pt (S.G.); cmanso@ufp.edu.pt (M.C.M.); 2Fernando Pessoa Energy, Environmental and Health Research Unit (FP-ENAS), 4249-004 Porto, Portugal

**Keywords:** dental bonding, humans, adhesives, tooth cervix, randomised controlled trial as topic, composite resins

## Abstract

This prospective, double-blind, six-arm parallel randomised controlled trial aimed to compare the performance of two universal adhesives (UAs) in non-carious cervical lesions (NCCLs), using the FDI criteria, and analysed if participants/NCCLs’ characteristics influenced the outcome. Thirty-eight 18- to 65-year-old participants were seeking routine dental care at a university clinic. At baseline, 210 NCCLs were randomly allocated to six groups (35 restorations’ each). The UAs tested were FuturabondU (FBU) and AdheseUniversal (ADU) applied in either etch-and-rinse (ER) and self-etch (SE) modes. FuturabondDC (FBDC) in SE and in SE with selective enamel etching (SE-EE) modes were controls. NCCLs were restored with AdmiraFusion. The analysis included nonparametric tests, Kaplan-Meier and log-rank tests (α = 0.05). At 2-years, of 191 restorations, ten were missed due to retention loss (all groups, *p* > 0.05). FBDC (*p* = 0.037) and FBU (*p* = 0.041) performed worse than ADU in SE mode. FBDC and FBU also showed worse functional success rate (*p* = 0.012, *p* = 0.007, respectively) and cumulative retention rates (*p* = 0.022, *p* = 0.012, respectively) than ADU. Some participants/NCCLs’ characteristics influenced (*p* < 0.05) the outcomes. FBU did not perform as well as ADU, especially in SE mode and due to functional properties. Participants’ age and NCCLs’ degree of dentin sclerosis and internal shape angle influenced FBU performance.

## 1. Introduction

Universal adhesives (UAs) are contemporary dental materials that can be applied alone by a simplified self-etch (SE) procedure or with phosphoric acid in selective enamel etching (SE-EE) or etch-and-rinse (ER) modes on dental hard tissues [1,2]. Clinicians must choose the suitable application protocol according to the dental hard tissues’ condition to optimize the final result of the procedure [3]. These universal solutions are similar to the prior one-step SE adhesives (1SAs) but with pH adjustment [4]. The addition of acidic functional monomers, such as 10-methacryloyloxydecyl dihydrogen phosphate (10-MDP), to UAs distinguishes them from the classical 1SA. That was the main change proposed by manufacturers to improve these materials. It is still challenged whether or not UAs are suitable for all adhesive procedures, since the durability and stability of the bonded interfaces also continues to be questionable [5,6].

Although UAs may bond chemically to various substrates, this bond stability seems to be susceptible to hydrolytic degradation and according to the adhesive applied [6]. Bonding a UA to enamel in SE mode may result in lower bond strength than in ER mode. On the other hand, in vitro studies have reported that only a prior etching enamel with phosphoric acid (SE-EE) results in significantly better bonding performance and durability [7,8]. On the other hand, it has been suggested, that the use of some UAs in dentin should not be preceded by phosphoric acid etching [9]. Bond strength to dentin of UAs dependents on their pH and bonding performance of the resultant adhesive interface stability is influenced by adhesives formulations [5,6,9]. Different co-monomers, solvents and catalysts might lead to variations in UAs’ film properties, reactivity, and bonding capacity regarding dentin [10].

“Non-carious cervical lesions (NCCLs) involves a cervical teeth wear, with loss of hard tissues unrelated to carious disease. Commonly, lesions are generated by multifactorial conditions related with changes in lifestyle and diet. Aging is also related with a higher teeth cervical wear. The clinical appearance of NCCLs can vary depending on the type and severity of the etiologic factors involved. Adhesive restoration and suitable control of etiological factors are appropriate clinical procedures to prevent the progression of those cavity lesions [11]. Composite restoration of NCCLs are challenging due to the adhesive reactions with enamel and various dentin tissues, the biomechanical aspects of the cervical area and, the limitations on the access and isolation of the operative field [12]. The impregnation of the dentine substrate by resin monomers and the stability of the bonded interface are of paramount importance to clinical performance [13]”.

Contrary to the amount of laboratory findings [6], not many randomised clinical trials (RCTs) evaluated the performance of UAs in NCCLs applied with different adhesion modes [2,14,15,16,17,18,19,20,21,22,23,24,25,26]. We performed the one-year recall of the present RCT [23]. Thirty-six participants and 199 restorations were examined, at that time. The UAs tested, FuturabondU (FBU) and AdheseUniversal (ADU), applied by both ER and SE modes, showed similar and acceptable clinical performance. Less satisfying marginal adaptation (*p* < 0.05) were registered when applied the 1SA (FuturabondDC; FBDC, control group), by both SE and SE-EE modes, than the UAs. The overall success rates (*p* > 0.05) were: 93.9% (FBDC-SE), 97.0% (FBDC-SE-EE, FBU-ER, FBU-SE) and 100.0% (ADU-ER, ADU-SE). Five (2.5%) restorations (2 with FBDC-SE = 2; and one with each FBDC-SE-EE, FBU-ER and FBU-SE) were missing due to retention lost (*p* > 0.05). The success and retention rates were similar and not dependent on materials or adhesion modes. The UAs revealed better clinical outputs than the 1SA (FBDC), particularly in SE mode.

Additionally, results of RCTs may be heterogeneous due to differences among cultures, foods, and habits [2,17,26,27]. The needs for clinical outcomes evidence of marketed UAs, already being applied in dental practice, and of subject’s and NCCLs’ characteristics that may influence the adhesive restorations clinical performance led to our study. As primary outcome of this RCT we assessed, using the FDI criteria, and compared for 2 years the overall clinical performance, success, and retention rates of two UAs both applied in SE and ER modes in NCCL restorations [28]. As secondary outcome, we analysed if the participants/NCCLs’ baseline characteristics influenced the primary clinical outcomes.

## 2. Materials and Methods

### 2.1. Trial Design

This prospective, double-blind, six-arm (two control groups) RCT followed the Consolidated Standards of Reporting Trials (CONSORT) statement [29]. The University Fernando Pessoa (UFP Ethics Committee, the National Ethics Committee for Clinical Research (20150305), and the National Authority of Medicines and Health Products (EC/011/2015) approved the research protocol. This clinical trial was registered at ClinicalTrials.gov (NCT02698371).

### 2.2. Sample Size

We calculated the sample size based on the primary outcome, using rules of thumb, considering the little or no information on the clinical performance of adhesion modes and all adhesives tested in this study. At the time this clinical trial was designed, we could not find viable information to allow calculating the sample size based on power analysis techniques. However, using rules of thumb for sample sizes, the overall sample required for a simpler comparative analysis using a McNemar test (repeated measures) with six groups, would need a total of at least 80 cases (teeth to restore). Consequently, each group needed at least 14 restorations. However, researchers assumed a cautious approach and increased considerably the minimum number of restorations in need stipulated in the aforementioned technique to a total of 35 restorations in each study group.

### 2.3. Participants

Eligible participants were 18- to 65-year-old subjects seeking routine dental care at our university clinic. An experienced clinician (P.M.M.) examined and screened participants who met the following inclusion criteria: the need to perform at least one and a maximum of six NCCLs deeper than 1.5 mm in both enamel and dentin tissues of vital premolars or molars. We excluded subjects that did not consent to participate, had less than 20 teeth in occlusion, were under orthodontic treatment, or were pregnant. Moreover, subjects with severe or chronic periodontal disease or periodontal surgery in the three previous months, with severe bruxism, with compromised medical, psychiatric, and pharmacotherapy history, with allergies and idiosyncratic reactions to the products, and with premolars or molar teeth that supported fixed/removal prosthesis were not included [23]. In our RCT we also considered as exclusion criteria subjects with poor oral hygiene or with high risk of caries or pulp injuries. The simplified oral hygiene index (OHI-s) was applied during recruitment, to include only subjects with very good or good oral hygiene. We considered subjects with “very good” and “good” oral hygiene those who presented OHI-S of 0.0–0.2 and of 1.3–3.0, respectively. Subjects who presented OHI-S of 3.1–6.0 were considered as having poor oral hygiene [30]. We used the International Detection and Assessment System II (ICDAS II) to assess “high risk of caries or pulp injuries”. We considered that subjects with at least one clinical risk factor, that was, subjects with recent caries experience (ICDAS II, scored different of 00) and presence of active caries lesion(s), as high risk of caries or pulp injuries. Subjects with low risk of caries, were those who had no clinical caries (ICDAS II, scored 00) risk factors [31]. During the recruitment, assessment and eligibility procedures, we did not find any subject classified as poor oral hygiene or high risk of caries or pulp injuries, so, we did not have to exclude subjects due to those clinical conditions.

### 2.4. Randomisation and Blinding

A randomization sequence was generated by the data analyst (M.C.M.) thanks to a block randomization table, where a permuted block of 35 restorations in each group was considered (two hundred and ten sealed envelopes were prepared, each with the information: adhesive system/adhesion mode for that restorative procedure). On the day of the restorative procedure, participants were allocated by the operator who enrolled them using sequentially numbered and sealed envelopes with the allocation cards previously prepared, opening sequentially as many envelopes as restorative procedures needed for that particular participant. Participants, examiners (J.D., L.T., S.G.) and data analyst (M.C.M.) were always blinded to group and NCCL teeth assignments during the study and none of them knew which treatment had been used, and participants could not note any differences. Due to the experience and clinical training the operator (P.M.M.) was not blinded to the application procedures.

### 2.5. Interventions

Three, nine, and twenty-six subjects received, respectively, three, five, and six NCCL restorations in one appointment. Before the restorative procedures, we collected the participant’s demographic (age, gender), behaviour (smoking habits and oral hygiene), and NCCL characteristics (premolar/molar tooth type, degree of dentin sclerosis [32], and internal angle shape cavity geometry [19]). We evaluated preoperative sensitivity by blowing air (10 s) using a dental syringe placed 2 cm from the tooth surface [19]. One experienced dentist (P.M.-M.) performed NCCLs restorations. No cavity preparation/design such as, enamel bevel, enamel/dentin roughened with diamond burns, or mechanical retentions were performed in NCCLs. A conservative dental hard tissues concept and the NCCLs design was considered as acceptable clinical conditions to achieve, possibly, the gold of adhesives restorations. After shade selection, all NCCLs cavities were properly cleaned with pumice and water in a rubber cup followed by rinsing and drying. All operative procedures were carried out under anaesthesia (Scandinibsa 3% mepivacaíne, Inibsa Dental S.L.U, Barcelona, Spain) and relative field isolation, with cotton rolls and retraction cord (Ultrapak #000 or #00, Ultradent Products, Inc., South Jordan, UT, USA). All single-dose adhesives were applied according to instructions (Table 1) and light-cured with a light emitting diode (LED Unit, Woodpecker; Guilin Woodpecker Medical Instruments Co., Lda, Guilin, China) 1000 mW/cm2 for 20 s. NCCLs were incrementally restored with Admira Fusion Ormofil (Voco, Cuxhaven, Germany). Each increment was light cured for 20 s, except the last one, which was light cured for 40 s. After removing the retraction cord, all restorations were finished and polished with diamond disks (OptiDisc medium and extra-fine course; Kerr Hawe SA; Bioggio, Switzerland) under water spray. Digital photographs were recorded. The UAs, Futurabond U (FBU) and Adhese Universal (ADU), were both applied in SE and in ER modes, as four testing groups. The 1SA, Futurabond DC (FBDC) applied by SE and by SE-EE modes, were the control groups. The materials and the procedures are detailed in Table 1.

### 2.6. Follow-Up Examinations and Outcomes

Three experienced and calibrated examiners (J.D., L.T., S.G.) evaluated all restorations at baseline and at the 1- and 2-year recalls, using FDI criteria [28]. We used the intra-class correlation coefficient (ICC) to calculate the intra-examiner (ICC ≥ 0.958) and inter-examiner agreement (ICC ≥ 0.952) at the beginning of the RCT. The examiners evaluated the restorations and the effect changes over time.

As recommendations for conducting controlled clinical trials [28], each examiner used a two-step approach for assigning scores for each parameter. First, each restoration was assessed to determine the level of clinical acceptability (Score 1, 2 or 3) or unacceptability (Score 4 and 5), for each parameter in each of the categories (aesthetic, functional, biological). Secondly, a further distinction was made, between an excellent (EX, score 1), Good (GO, score 2), and Sufficient (SS, score 3) for each aesthetic (staining margin), functional (fractures/retention and marginal adaptation) and biological (postoperative (hyper-) sensitivity and caries recurrence) property. The restorations clinically unacceptable were scored as 4, Unsatisfactory (UNS; needed repair for prophylactic reasons) or as 5, Poor (PO; needed replacement).

The score 1 (EX) indicated that the quality of the restoration was excellent/fulfilled all quality criteria, and the tooth and/or surrounding tissues were adequately protected. The score 2 (GO) was registered when the quality of the restoration was still highly acceptable, though one or more criteria deviated from the ideal. The restoration could be modified by polishing and upgraded to an ‘excellent’ rating, but this was not normally necessary. There was no risk of damage to the tooth and/or the surrounding tissues. The Score 3 (SS) indicated that the quality of the restoration was sufficiently acceptable but with minor shortcomings. Because of their location/extent, however these could not be eliminated without damage to the tooth, though no adverse effects were anticipated. The restorations that scored 4 were UNS, but repaired whereas, those that scored 5 (PO) were replaced [28].

We considered restorations with a decrease in overall clinical acceptance from scores Excellent (EX) to good (GO) or Sufficient (SS) as having acceptable/sufficient clinical performance. An insufficient or low performance corresponded to restorations scored as unacceptable, needing repair for prophylactic reasons (UNS), and poor (PO). We calculated restoration success rates as the percentage of restorations classified with acceptable esthetic, functional, and biological properties (scores EX, GO, SS). The retention rate corresponded to the percentage of restorations not missing due to fracture and lost retention, considering the number of restored teeth available for observation at each recall.

### 2.7. Statistical Analysis

We conducted the statistical analyses using the IBM SPSS Statistics version 24 software (IBM Corp, Armonk, NY, USA), considering a significance level of 0.05 for all statistical inference situations. We calculated the intra- and inter-examiner agreement through the ICC. We compared categorical variables per group using the chi-square test and used the McNemar or the Wilcoxon tests for the pairwise longitudinal comparison (baseline up to 2-year recall) of categorical ordinal variables for each group’s esthetic, functional, and biological properties. We conducted pairwise comparison (performance, success, and retention rates) from baseline up to the 2-year follow-up between research groups, SE and SE-EE/ER adhesion modes, and adhesives with Kaplan-Meier survival curves and log-rank tests. Finally, we used the Mantel-Cox log-rank test to compute the overall clinical acceptance of restorations at the 2-year follow-up analysis, according to participants/NCCLs’ baseline characteristics.

## 3. Results

Of the baseline 210 NCCL restorations’ (38 participants included) we observed 191 restorations (35 participants) at the 2-year recall. Five restorations had been lost due to retention at the 1-year recall and five others at the 2-year recall (Figure 1). We found no differences regarding participants’ and NCCLs’ characteristics (Table 2) at baseline.

### 3.1. Esthetic, Functional, and Biological Properties

Table 3 summarizes the clinical evaluation of NCCL restorations by FDI criteria during the 2-year follow-up. We found no differences in any group regarding staining margin at the 2-year follow-up. Fractures and retention clinical acceptance decreased in FBU-SE (*p* = 0.041) at the 2-year recall. Marginal adaptation decreased in FBDC-SE (*p* = 0.009), in FBU (ER; *p* = 0.038 and SE; *p* = 0.024) and in ADU (ER; *p* = 0.038 and SE; *p* = 0.023) at the 2-year recall. Regarding biological properties, postoperative (hyper-)sensitivity and caries recurrence showed no significant deviations (McNemar; *p* > 0.05) in any group during the 2 years.

### 3.2. Clinical Performance

The cumulative decrease in clinical performance with the SE adhesion mode (Table 4) was acceptable in FBDC-SE (*p* = 0.037) and in FBU-SE (*p* = 0.041) but both worse than in ADU-SE. We detected no differences between FBDC-SE and FBU-SE, nor between the same UAs applied in SE-EE and in ER adhesion modes, respectively. Pairwise comparison of the adhesives revealed that FBDC (*p* = 0.027) and FBU (*p* = 0.029) had acceptable clinical performance but worse than ADU from baseline to the 2-year recall.

### 3.3. Success Rates

The overall success rates (log-rank test, *p* ≥ 0.037) were 87.5% for FBDC-SE, 90.6% for FBDC-SE-EE and for FBU-ER, 87.9% for FBU-SE, 96.9% for ADU-ER, and 100% for ADU-SE. The overall success rate was significantly lower for FBDC-SE and FBU-SE compared to ADU-SE (*p* = 0.037 and *p* = 0.041, respectively).

The cumulative decrease of functional success rates’ revealed lower properties in the FBDC (91.4% in SE and 91.2% in SE-EE modes’) and in the FBU (88.6% in SE and 91.4% in ER modes’) groups’ compared to the ADU (100.0%) groups (*p* = 0.012 and *p* = 0.007, respectively, Table 4) The FBU-SE (88.6%) had lower (*p* = 0.040) cumulative functional success rate than ADU-SE (100%). The esthetic and biological properties not differed (log-rank test, *p* ≥ 0.05) over time.

### 3.4. Retention Rates

The overall retention rate at 2 years was 94.8%. The cumulative decrease in retention rates (Table 4) was similar (*p* ≥ 0.074) among groups for all adhesion modes, but the FBDC and FBU groups had lower retention rates than ADU. We detected no differences between FBDC and FBU.

In this clinical research, the annual failure rates showed significant higher values for FBDC-SE (12.5%) and FBU-SE (12.1%) than for ADU-SE (0%) (*p* = 0.037 and *p* = 0.041, respectively), while no significant differences were accounted for the other groups, with 9.4% for FBDC-SE-EE and FBU-ER, and 3.1% for ADU-ER. At the 2-year follow-up, 91.4% (all groups excepted ADU-SE) and 94.3% (ADU-SE) of the sample were evaluated.

### 3.5. Participants’ and NCCLs’ Characteristics

We analysed the cumulative clinical performance according to the participants’ and NCCLs’ baseline characteristics (Table 4). The median age of the participants was defined as the cut-off for the comparisons. Participants aged >56 years old showed worse performance of FBU adhesive restorations. We also found an association between NCCLs’ dentin sclerotic degree and worse performance with the SE adhesion mode and lower retention rates with the SE adhesion mode and FBU adhesives. NCCLs’ internal shape angle was associated with worse performance with the SE-ER/ER adhesion modes and FBU adhesive restorations. We detected no differences in the FBDC and ADU adhesives.

## 4. Discussion

This RCT aimed to compare the performance of two universal adhesives (UAs) in non-carious cervical lesions (NCCLs) using the FDI criteria and analyse if participants/NCCLs’ characteristics influenced the outcome. Thirty-eight participants, 21 male and 17 female, with a median age of 56 years (range 24–63 years), received 210 NCCL restorations. Other UA clinical trials enrolled a similar number of participants (20 to 55) [2,14,17,19,21,22,24,25,26,33].

In the literature, eight RCTs evaluated the UA Scotchbond Universal in SE and ER modes at different recalls (6 to 36 months) [2,15,17,19,22,25,26,34]. Other RCTs tested the UAs Xeno Select [24], FBU [12,18], Prime & Bond Elect [14], Tetric N-Bond Universal (ADU in some countries) [12,33], Gluma Universal [21], ClearfilT Universal Bond, iBond Universal, and G-Premio BondTM [20], with 6- to 24-month recalls. Our RCT, after the 1-year recall [23], assessed and compared the performance of two UAs—FBU and ADU—from different manufacturers. We chose FBDC for the control group because it was the last generation of 1SA from the same manufacturer of FBU. Only Morsy and colleagues’ clinical trial also compared the 1-year performance of FBU and ADU [12]. We could have designed this RCT using as control group an adhesive more studied and with a more reliable behaviour, such as the three-step ER adhesive Scotchbond Multipurpose Plus and/or the two-step SE adhesive Clearfil SE Bond, considered as gold standards [9]. However, as stated by some authors, although RCTs with long-term follow-ups are the best study designs to evaluate clinical performance of adhesives, their validity relies heavily on rigorous study design and completeness of subject follow-up [6]. Many adhesives are commercially available on the market for clinical practice, with no or very few evidence regarding clinical performance. Additionally, frequent introduction of new dental adhesives quickly outdates existing products. This often tempts manufacturers to release successor products even before its precursor has been clinically evaluated, as in case of the FBDC and the FBU adhesives. It can be also hypothesized that formulation of new adhesives and its precursor, of the same manufacture, could have similar chemical (monomers purity and concentration) backgrounds, that can influence adhesive´s bond strength and bonding stability over time. Currently, SE modalities with adhesives including monomers, such as FBU [35] and ADU (Table 1), which include the 10-MDP, are considered the most reliable option for dentin [4,36,37,38], to improve the long-term durability of both dentin and enamel bonding [4,5,9,39,40]. ADU additionally have another functional monomer, the methacrylated carboxylic acid polymer (MCAP). However, bond stability of resin–dentine interfaces created by current UAs containing 10-MDP could be doubtful in clinical scenarios, since nano-layering by 10-MDP-Ca salts may not be so invulnerable to degradation as the manufacturers would like [41,42]. Like other UAs, FBU and ADU contains 2-hydroxyethyl methacrylate (HEMA) to regulate hydrophilicity and also some water contents [25,26,43] but are more hydrophobic, due to acidic monomers and MDP higher contents, than previous generations of 1SA (highly hydrophilic) [44]. Furthermore, in some UAs, the higher proportion of water content (to ionize the acidic monomers and demineralize dentin and enamel) might lead to interface degradation from the residual water left in the cured adhesive layer, as recently reported (3% in ADU to 10–15% in Scotchbond Universal, and up to the 25% in the UA G-Premio Bond) [45]. Those conditions of UAs may explain differences in the functional effects (retention and marginal integrity) observed at our 2-year recall. Thus, UAs’ clinical performance may be highly product-dependent [46,47]. The adhesive´s monomers conversion could be affected by light-curing procedures. Moreover, the pH of some products could be disturbed by the acidic conditions. Both aspects variations had potential effects on the interface bonding stability and long-term functional properties of adhesive restorations [9].

We found no differences in any group regarding staining margin, postoperative (hyper-) sensitivity, and recurrence of caries at the 2-year recall, which is in agreement with the main findings of RCTs focused on UAs [18,19,22,24,25,33]. Fractures and retention clinical acceptance significantly decreased in FBDC-SE at the 1-year recall [23] and in FBU-SE at the 2-year recall. Marginal adaptation decreased in FBDC-SE at the 1-year [23] and the 2-year recalls, but with the UAs (with both, ER and SE mode), the FBU and the ADU this occurrence were significantly registered, only at the 2-year recall. We detected a decrease in marginal adaptation in the control group FBDC-SE and both the UAs with the SE and ER modes at the 2-year recall, but with similar outputs among FBU and ADU adhesives. Morsy and colleagues also found no differences between the FBU and ADU tested groups for marginal adaptation and discoloration and postoperative sensitivity [12]. Previous clinical research supports these findings [12,18,19,26,33,34].

We observed a decrease in overall clinical acceptance due to fractures and retention for FBU-SE at the 2-year follow-up, but no differences among the other UA groups. Morsy and colleagues’ clinical trial reported 100% retention rates for FBU and ADU at the 1-year recall [12]. Loguercio and colleagues reported 15 restorations lost with the Tetric N-Bond Universal (namely, ADU), with no differences between any pair of groups, at the 18-month recall [33]. In our study, the cumulative decrease in retention rates was similar between the SE and SE-EE adhesion modes. Fracture and loss of retention are clinical events and “hard criteria” of evaluation. The diagnosis of lost restoration is clear, usually evident to subject and requires a replaced restoration. In our study, previously lost retention restorations in all cases were missing, were scored as poor, and therefore were replaced. At one-year recall, one restoration with FBDC-SE and other with ADU-ER scored as unsatisfactory for fractures and retention (Table 3) and were clinically repaired. In our study, all the restorations that scored as clinically unsatisfactory for FDI criteria at the first year recall were accounted as cumulative failure, at the 2-years follow-up analysis. It could be considered that repaired restorations were therefore scored as “relative failure” and replaced restorations as “absolute failure”. As expressed by Hickel and colleagues, most frequently, FDI score 5 shows worse clinical events than score 4, but that is not inevitable. Score 4 and the possibility for restoration repair depends more on the location and size of the defect, as occurred in the present study, and therefore it must be decided whether the restoration could be repaired or requires replacement [48]. Some RCTs compared retention and absence of fractures with several adhesion modes and found similar outputs [18,19,22,24,25,33], though some suggested, or statistically showed, that UAs in ER or SE-EE modes promoted better retention results [18,19,22,24,25]. However, and irrespectively of the adhesion mode, in our study, FBDC and FBU showed acceptable but lower retention rates than ADU during the 2 years of follow-up. Furthermore, FBDC-SE and FBU-SE had acceptable but worse performance than ADU-SE, and FBDC and FBU showed acceptable but worse performance, functional success, and cumulative retention rates than ADU at the 2-year follow-up.

Only a few studies analysed the influence of participants/NCCLs’ characteristics on the clinical performance of restorations bonded with UAs [21,49]. Within this study, restoration overall decreases in clinical performance of NCCLs with FBU increased with age (over 56 years). Age-related changes in the structure of teeth and the cavity size may explain those effects already reported in the literature [50]. Considering that 90% of the tooth tissue in NCCLs is sclerotic dentin, resin hybridization could be difficult [14,51,52,53], and some of our secondary outputs might be due to insufficient mechanical and chemical conditioning of the sclerotic dentin [1,4]. The dentin roughening before clinical UAs application had been suggested in literature but results are scarce and some controversial [33]. When using SE modes with ultra-mild UAs, such as the FBU and the ADU, the micro-retentive bonding to this substrate depends mainly on chemical bonding. However, additional etching of the dentin with phosphoric acid may lead to excessive demineralization, and collagen hydrolysis may occur on the bottom of the hybrid layer [54]. In our perspective, the UA’s formulations details also need to be more fully described by manufacturers’ labels.

Our study design is in agreement with other UA clinical studies [2,14,17,19,21,22,24,25,26,33]. However, the reliability of the secondary outputs’ statistical analysis may represent a limitation because of the very small group size, which is quite important when discussing the results regarding the possible influence of NCCL restorations with dentin sclerosis degrees 3 or 4, or those with an obtuse internal shape angle. Within our analysis, the log- rank test was used to compare the time to restoration failure with the “levels” of each participant/NCCLs characteristics. To have reasonable sample size in each characteristic, some recoding of those factors was required, as for participants-age. For the dentin sclerosis and the internal shape angle (factors’ with more than two levels) multiple comparisons were done to try to identify which levels were significant with regard to differences in time to failure. This pairwise multiple comparison had an exploratory purpose but with the limitation of the sample size. However, in future analyses, regression models should be applied in order to try to identify if combination of all those participants/NCCLs characteristics really have significant effects on the time of restorations failure. Another limitation of this RCT is its duration (2-years evaluation). Future trials controlling these aspects and including longer-term data might provide more accurate outcomes, and identifying the risk factors, using statistical inference techniques.

## 5. Conclusions

At the 2-year follow-up, all adhesives revealed acceptable clinical performance. ADU showed better clinical performance, functional success (marginal adaptation), and retention rates than FBU and the control FBDC, especially in the SE mode. FBU-SE showed lower functional properties than ADU-SE. Participants’ age and NCCLs’ degree of dentin sclerosis and internal shape angle influenced the UA FBU and the adhesion modes´ cumulative clinical outputs; thus, further investigations should consider both the time to restoration failure and the multiple comparation of participants’/NCCLs’ characteristics as factors that may induce the clinical events.

## Figures and Tables

**Figure 1 polymers-14-00033-f001:**
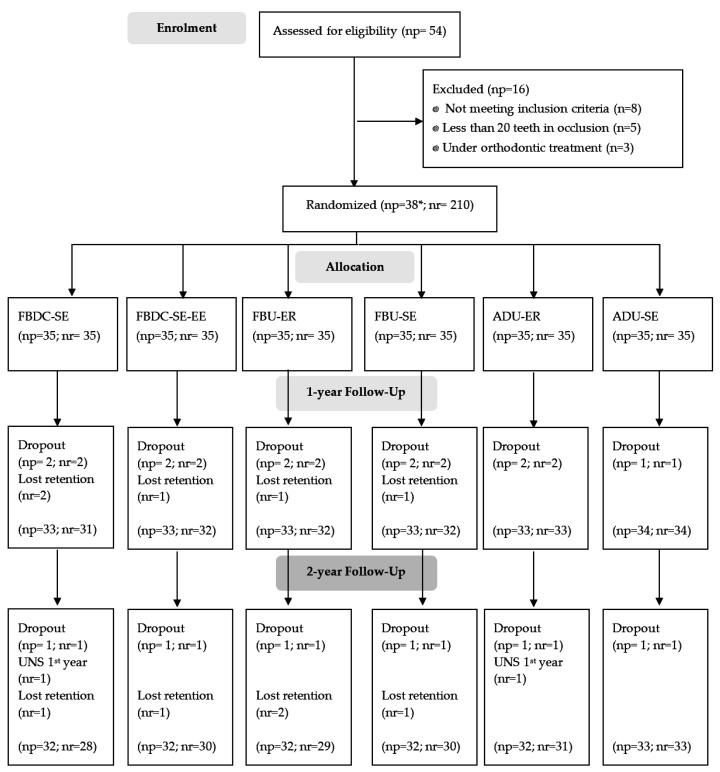
Research Flow chart of participants/NCCLs enrolled, followed, and analysed. np: number of participants; nr: number of restorations; SE: self-etch. SE-EE: self-etch with selective enamel etching; ER: etch-and-rinse; FBDC: Futurabond DC; FBU: Futurabond U; ADU: Adhese Universal; UNS: scored clinically unsatisfactory. * Each participant was enrolled in several arms and could receive at least 1 up to 6 NCCLs restorations.

**Table 1 polymers-14-00033-t001:** Details of the materials used in each research group and their application procedures.

Material (Manufacturer) Lot # Number	Composition
Futurabond DC (FBDC) (Voco, Cuxhaven, Germany) Lot# 1532592	Liquid 1. Acidic adhesive monomer ^1^; Bis-GMA (5–10%), HEMA (5–10%) ;Liquid 2. Ethanol (50–100%); Initiator (2.5–5%) Mixture. organic acids, BIS-GMA, HEMA, TMPTMA§, camphorquinone, amines (DABE), BHT, catalysts, fluorides, and ethanol pH = 1.5
Futurabond^®^ U (FBU) (Voco, Cuxhaven, Germany) Lot# 1543141	Liquid 1. (2-HEMA) (25–50%); BIS-GMA (25–50%); HEDMA (10–25%); Acidic adhesive monomer (5–10%); UDMA (5–10%); catalysts (≤2.5%), silica nanoparticles; Liquid 2. Ethanol (50–100%); Initiator (2.5–5%); catalysts (≤2.5) pH = 2.3
Vococid (Voco, Cuxhaven, Germany) Lot# 152135	35% orthophosphoric acid, water, synthetic amorphous silica, polyethylene glycol, aluminum oxide
Adhese Universal (ADU) (Ivoclar Vivadent AG,Liechtenstein) Lot# U35131	Liquid: 2-HEMA (10 -< 25%); Bis-GMA (10 -< 25%); ethanol (10 -< 25%); 1,10-decandiol dimethacrylate (3 -< 10%); Methacrylated phosphoric acid ester (3 -< 10%); camphorquinone (1 -< 2.5%); 2-dimethylaminoethyl methacrylate (1 -<2.5%); 2-dimethylaminoethyl methacrylate (0.1 -< 2.5%). pH = 2.5–3.0
Admira Fusion (Voco, Cuxhaven, Germany) Lot# (Shade A1, A2, A3, A3,5) 1508270, 150827, 1510508, 1509381	Ormocer composite resin (organically modified ceramics, according to the respective manufacturer); camphorquinone, amine, BHT, classical silica particles (20–40 nm), Ba-Al-Si glass (1 μm), iron oxide, titanium dioxide
**Research groups:** **Adhesive-adhesion mode**	**Application Procedures**
FBDC-SE (Control group)	Mixture Liquid 1 into Liquid 2 (1:1 ratio). Apply and rub this homogeneous mixture to enamel and dentine for 20 s; Air-blow for 5 s; light cure (1000 mW/cm^2^) for 20 s.
FBDC-SE-EE (Control group)	Apply etchant selectively on enamel and leave for 30 s. Thoroughly rinse for 1 min and gently dry. Dentine surface must remain slightly wet. Mixture Liquid 1 into Liquid 2 (1:1 ratio). Apply and rub this homogeneous mixture to enamel and dentine for 20 s; Air-blow for 5 s; light cure (1000 mW/cm^2^) for 20 s.
FBU-ER	Apply etchant for 30 s on enamel and 15 s on dentine; Thoroughly rinse for 1 min and gently dry. Dentine surface remains with a silky matt appearance. Apply and rub adhesive for 20 s, and air-blow for 5 s; light-cured (1000 mW/cm^2^) for 20 s.
FBU-SE	Apply and rub adhesive for 20 s, and air-blow for 5 s; light-cured (1000 mW/cm^2^) for 20 s.
ADU-ER	Apply etchant for 30 s on enamel and 15 s on dentine; Thoroughly rinse for 1 min and gently dry. Dentine surface remains dry. Scrubbed adhesive for at least 20 s; Air-blow to disperse adhesive until a glossy, immobile film layer result; Light-cure (1000 mW/cm^2^) for 20 s.
ADU-SE	Scrubbed adhesive for at least 20 s; Air-blow to disperse adhesive until a glossy, immobile film layer result; Light-cure (1000 mW/cm^2^) for 20 s.

^1^ 10-MDP—10-methacryloyloxydecyl dihydrogen phosphate. Bis-GMA-bisphenol A glycidyl methacrylate. HEMA-2-hydroxyethyl methacrylate. TMPTMA—Trimethylolpropane trimethacrylate. DABE—N,N-dimethyl-p-aminobenzoic acid ethyl ester. BHT- Butylated hydroxytoluene. HEDMA—hydroxyethyl dimethacrylate. UDMA—Urethane dimethacrylate. SE: Self-etch; SE-EE. Self-etch with selective enamel etching. ER: Etch-and-rinse.

**Table 2 polymers-14-00033-t002:** All participants’ and NCCL restorations’ characteristics, counts (n), and percentages (%) at baseline, allocated to the control (FBDC-SE; FBDC-SE-EE) and study (FBU and ADU) groups.

Participants/NCCLs’ Characteristicsat Baseline		ALL	FBDC SE	FBDC SE-EE	FBU ER	FBU SE	ADU ER	ADU SE	*p* **
Age	n	38	35	35	35	35	35	35	
	Me (P_25_–P_75_)	56 (41–59)	56 (41–60)	56 (43–59)	55 (41–58)	56 (43–59)	56 (40–60)	56 (41–60)	0.999
	min–max	24–63	24–63	24–63	24–63	24–63	24–63	24–63	
Gender	Female	17 (44.7%)	15 (42.9%)	16 (45.7%)	16 (45.7%)	16 (45.7%)	14 (40%)	14 (40%)	0.990
	Male	21 (55.3%)	20 (57.1%)	19 (54.3%)	19 (54.3%)	19 (54.3%)	21 (60%)	21 (60%)	
Smoking habits	No	32 (84.2%)	30 (85.7%)	30 (85.7%)	30 (85.7%)	30 (85.7%)	29 (82.9%)	30 (85.7%)	0.999
	Yes	6 (15.8%)	5 (14.3%)	5 (14.3%)	5 (14.3%)	5 (14.3%)	6 (17.1%)	5 (14.3%)	
Oral hygiene [30]	Very Good	25 (65.8%)	23 (65.7%)	25 (71.4%)	25 (71.4%)	24 (68.6%)	23 (65.7%)	22 (62.9%)	0.966
	Good	13 (34.2%)	12 (34.3%)	10 (28.6%)	10 (28.6%)	11 (31.4%)	12 (34.3%)	13 (37.1%)	
Number of cigarettes for smokers	n	6	5	5	5	5	6	5	
	Me (P_25_–P_75_)	14 (5.3–20)	15 (9.5–20)	13 (4.5–17.5)	13 (4.5–17.5)	13 (4.5–17.5)	14 (5.3–20)	15 (4.5–20)	0.943
	min-max	3–20	6–20	3–20	3–20	3–20	3–20	3–20	
Tooth type, n (%)									
Pre-molar tooth		176 (83.8%)	29 (82.9%)	32 (91.4%)	32 (91.4%)	27 (77.1%)	30 (85.7%)	26 (74.3%)	0.252
Molar tooth		34 (16.2%)	6 (17.1%)	3 (8.6%)	3 (8.6%)	8 (22.9%)	5 (14.3%)	9 (25.7%)	
Degree of dentin sclerosis [32] n (%) *									
Degree 1		146 (69.5%)	29 (82.9%)	24 (68.6%)	26 (74.3%)	20 (57.1%)	23 (65.7%)	24 (68.6%)	0.353
Degree 2		35 (16.7%)	4 (11.4%)	7 (20%)	5 (14.3%)	7 (20%)	5 (14.3%)	7 (20%)	
Degree 3,		8 (3.8%)	0 (0%)	1 (2.9%)	0 (0%)	3 (8.6%)	4 (11.4%)	0 (0%)	
Degree 4		21 (10%)	2 (5.7%)	3 (8.6%)	4 (11.4%)	5 (14.3%)	3 (8.6%)	4 (11.4%)	
Cavity geometry (internal shape angle, °) [19] n(%)									
Acute (<45°)		84 (40%)	13 (37.1%)	17 (48.6%)	14 (40%)	14 (40%)	15 (42.9%)	11 (31.4%)	0.903
Acute-to-Right (45–90°)		60 (28.6%)	9 (25.7%)	11 (31.4%)	11 (31.4%)	9 (25.7%)	8 (22.9%)	12 (34.3%)	
Obtuse (>90°)		66 (31.4%)	13 (37.1%)	7 (20%)	10 (28.6%)	12 (34.3%)	12 (34.3%)	12 (34.3%)	

Me: Median. * Degree 1: No sclerosis present. Dentine is light yellow or whitish with little discoloration; Dentine is opaque, with little translucency or transparency. Degree 2: More than category 1, but less than 50% of the difference between categories 1 and 4. Degree 3: Less than category 4, but more than 50% of the difference between categories 1 and 4. Degree 4: Significant sclerosis present. Dentine is dark yellow or even discolored (brownish). Dentine appears glassy, with significant translucency or transparency evident. **According to the chi-square test.

**Table 3 polymers-14-00033-t003:** Clinical evaluation by FDI criteria of NCCL restorations’ distribution (number) per group during the 2-year follow-up [28].

FDI Criteria/Score		Restorations (n) at Baseline and 2-Year Follow-Up											
		FBDC-SE		FBDC-SE-EE		FBU-ER		FBU-SE		ADU-ER		ADU-SE	
		Base	2y	Base	2y	Base	2y	Base	2y	Base	2y	Base	2y
Staining margin	EX	35	26	35	29	35	28	35	28	35	29	35	30
	GO	-	2	-	-	-	-	-	1	-	-	-	1
	SS	-	-	-	1	-	1	-	1	-	1	-	2
	UNS	-	-	-	-	-	-	-	-	-	1	-	-
Fractures and Retention	EX	35	28	35	29	35	29	35	27 *	35	31	35	32
	GO	-	-	-	-	-	-	-	2 *	-	-	-	1
	SS	-	-	-	-	-	-	-	-	-	-	-	-
	UNS	-	-	-	1	-	-	-	1 *	-	-	-	-
	PO	-	1	-	1	-	2	-	1 *	-	-	-	-
Marginal Adaptation	EX	31	20 *	33	29	33	24 *	28	24 *	32	26 *	33	27 *
	GO	4	5 *	2	-	2	3 *	7	3 *	3	3 *	2	2 *
	SS	-	3 *	-	1	-	2 *	-	3 *	-	2 *	-	4 *
	UNS	-		-	-	-	-	-	-	-	-	-	-
Postoperative (Hiper-) sensitivity	EX	35	28	35	28	35	29	35	28	34	30	34	32
	GO	-	-	-	2	-	-	-	2	1	1	1	1
Recurrence of Caries	EX	35	28	35	29	35	29	35	29	35	31	35	33
	GO	-	-	-	-	-	-	-	1	-	-	-	-
	SS	-	-	-	-	-	-	-	-	-	-	-	-
	UNS	-	-	-	1	-	-	-	-	-	-	-	-

EX: clinically excellent/very good; GO: clinically good; SS: clinically sufficient/satisfactory; UNS: clinically unsatisfactory (repair for prophylactic reasons); PO: clinically poor (replacement necessary). Base: Baseline. y: year. * *p* < 0.05 according to Wilcoxon or McNemar tests, i.e., significant differences from baseline to the 1-y and 2-y recalls.

**Table 4 polymers-14-00033-t004:** P value of the NCCLs group’ restorations pairwise comparisons (log-rank test, Mantel-Cox) for the clinical performance, clinical properties success and retention rates, and participants/NCCLs’ baseline features comparison regarding the overall clinical performance and the retention rates, by FDI criteria, at the 2-year recall [28].

Primary Outcome NCCLrestorations’ Performance		SE Mode			SE-EE/ER Modes			Adhesives		
		FBDC vs. FBU	FBDC vs. ADU	FBU vs. ADU	FBDC vs. FBU	FBDC vs. ADU	FBU vs. ADU	FBDC vs. FBU	FBDC vs. ADU	FBU vs. ADU
Overall Clinical performance		0.925	**0.037**	**0.041**	1.000	0.300	0.300	0.947	**0.027**	**0.029**
Esthetic success rate		1.000	1.000	1.000	1.000	0.325	1.000	1.000	0.333	0.325
Functional success rate		0.767	0.074	**0.040**	1.000	0.078	1.000	0.791	**0.012**	**0.007**
Biological success rate		1.000	1.000	1.000	0.317	0.310	1.000	0.310	0.302	1.000
Retention rate		0.953	0.074	0.079	0.650	0.154	0.078	0.791	**0.022**	**0.012**
Secondary Outcome Participants/NCCLs characteristics		SE mode			SE-EE/ER modes			FBDC	FBU	ADU
**Age**	≤56 vs. >56 y	0.129			0.752			0.204	**0.023**	0.191
**Gender**	Fem. Vs. Masc.	0.831			0.479			0.463	0.948	0.429
**Smoking Habits**	No vs. Yes	0.965			0.283			0.859	0.303	0.670
**Oral Hygiene [30]**	Very Good vs. Good	0.965			0.283			0.610	0.846	0.206
**Tooth type**	Pre-molar vs. Molar	0.857			0.383			0.966	0.914	0.634
**Dentin Sclerosis** [32]	Degree * 1 vs. Degree 2	0.166			0.244			0.861	0.349	0.597
	Degree 1 vs. Degree 3	**0.021**			0.571			0.736	0.051	0.792
	Degree 1 vs. Degree 4	0.408			0.237			0.569	0.713	0.687
	Degree 2 vs. Degree 3	0.276			1.000			0.763	0.305	1.000
	Degree 2 vs. Degree 4	0.160			0.060			0.554	0.725	1.000
	Degree 3 vs. Degree 4	**0.019**			0.352			0.655	0.219	1.000
**Cavity geometry (internal shape angle)** [19]	Acute vs. Acute-to-Right	0.619			0.152			0.781	0.113	1.000
	Acute vs. Obtuse	0.272			0.284			0.155	0.476	0.317
	Acute-to-Right vs. Obtuse	0.137			**0.033**			0.143	**0.037**	0.363
**Retention Rate**										
**Age**	≤56 vs. >56 y	0.135			0.479			0.406	0.071	1.000
**Gender**	Fem. vs. Masc.	0.717			0.961			0.966	0.599	1.000
**Smoking Habits**	No vs. Yes	0.331			0.372			0.392	0.345	1.000
**Oral Hygiene**	Very Good vs. Good	0.485			0.702			0.196	0.933	1.000
**Tooth type**	Pre-molar vs. Molar	0.748			0.468			0.662	0.282	1.000
**Dentin Sclerosis**	Degree 1 vs. Degree 2	0.299			0.305			0.954	0.902	1.000
	Degree 1 vs. Degree 3	**0.007**			0.618			0.769	**0.049**	1.000
	Degree 1 vs. Degree 4	0.484			0.658			0.511	0.705	1.000
	Degree 2 vs. Degree 3	0.154			1.000			0.763	0.133	1.000
	Degree 2 vs. Degree 4	0.260			0.192			0.500	0.219	1.000
	Deg 3 vs. Deg 4	**0.019**			0.527			1.000	0.219	1.000
**Cavity geometry (internal shape angle)**	Acute vs. Acute-to-Right	0.643			0.151			0.775	0.116	1.000
	Acute vs. Obtuse	0.681			0.932			0.659	0.759	1.000
	Acute-to-Right vs. Obtuse	0.394			0.139			0.508	0.075	1.000

* Degree 1: No sclerosis present. Dentine is light yellow or whitish with little discoloration; Dentine is opaque, with little translucency or transparency. Degree 2: More than category 1, but less than 50% of the difference between categories 1 and 4. Degree 3: Less than category 4, but more than 50% of the difference between categories 1 and 4. Degree 4: Significant sclerosis present. Dentine is dark yellow or even discolored (brownish). Dentine appears glassy, with significant translucency or transparency evident.

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
