# Peer review of "Universal Adhesives and Adhesion Modes in Non-Carious Cervical Restorations: 2-Year Randomised Clinical Trial"

_polymers, 2021, doi:10.3390/polym14010033_

Round 1

Reviewer 1 Report

The authors present the 2-year results of a randomized clinical trial, reporting the performance of two universal adhesives applied in self-etch or etch-and-rinse modalities. The study protocol is described, along with the restorations clinical results and patient´s characteristics. Comparisons between the study groups are made and regarding the participant´s and the restored lesions characteristics.

Lines 23-24: the sentence “…was less likely to perform better…” is confusing. Please clarify the sentence meaning.

Line 53: please write the term in full before using the abbreviation NCCLs.

Line 60-61: I suggest this sentence to be removed since it is stated in the materials and methods section.

Since this is the 2-year report and the one-year report was already published, I suggest that the one-year results be added to the introduction section.

Line 68: since the authors refer to the use of CONSORT guidelines in study planning and reporting, the CONSORT checklist should be added as a supplementary file.

Section 2.2, lines 79-84: please briefly describe how the parameters were evaluated, for instance, good/bad oral hygiene, high risk of caries or pulp injuries… I saw the one-year publication, and it’s not described there.

Line 84: the number of included subjects is a result and should be moved to the appropriate section.

Section 2.3: were the cavities prepared in any way? Was sclerotic dentin partially removed to improve the adhesion?

Section 2.4: please describe how/what was accessed to attribute the scores of excellent, good, sufficient… in each category (esthetic, functional and biological). This part is unclear. For instance, the authors refer that the staining of the margins was evaluated for the esthetic property. If staining was present, which score was attributed? Or it was dependent on the staining intensity?

Figure 1: I suggest this figure be moved to the results section since it presents results from the 2-years follow-up, including the dropouts and the restorations clinical evaluation.

Section 2.5: if no data existed to perform a sample calculation, did the authors consider doing first a pilot study to acquire the necessary data to calculate the sample?

I suggest the materials and methods section be reorganized. For instance, the sample size determination is described after the patient’s enrolment and follow-up examinations. I recommend an organization corresponding to the study prosecution be adopted.

Table 2: what is the meaning of Me? Median? Please add it to the table caption.

Section 3.1 and table 3: the authors refer to one-year follow-up results that have been already published. The results from the one-year evaluation are not results from the present study and should not be included in the results section. Besides, no reference is made to the previous article, so this can be seen as plagiarism. You can refer to them in the introduction or discussion sections with the appropriate citation.

Table 4 and section 3.5: how was the age of 56 determined as the cutoff to comparisons?

Lines 315-317: the authors stated that “the restorations that scored as clinically unsatisfactory at the first-year recall were no longer included in the study”. This sentence is unclear. Does this mean they were excluded from the analysis and not accounted for? Or that they were not clinically replaced?

Author Response

Response to Reviewer 1 Comments

Manuscript ID- Polymers - 1489450

Comments and Suggestions for Authors. The authors present the 2-year results of a randomized clinical trial, reporting the performance of two universal adhesives applied in self-etch or etch-and-rinse modalities. The study protocol is described, along with the restorations clinical results and patient´s characteristics. Comparisons between the study groups are made and regarding the participant´s and the restored lesions characteristics.

Lines 23-24: the sentence “…was less likely to perform better…” is confusing. Please clarify the sentence meaning.

Answer: The part of that sentence was rewritten as “FBU did not perform as well as ADU, …

Line 53: please write the term in full before using the abbreviation NCCLs.

Answer: The term was described as indicated by the reviewer. Authors added in introduction a small previous paragraph on etiology and main adhesion problems related to NCCLs restorations, as suggested by reviewer 2.

Line 60-61: I suggest this sentence to be removed since it is stated in the materials and methods section.

Answer: We performed as suggested and removed the sentence “ A 1SA applied in SE and SE-EE modes was defined as control”, the line 60-61.

Since this is the 2-year report and the one-year report was already published, I suggest that the one-year results be added to the introduction section.

Answer: We added to the introduction section, a brief paragraph of the previous one-year report already published, as suggested by the reviewer:

We performed the one-year recall of the present RCT [23]. Thirty six participants and 199 restorations were examined at that time. The UAs tested – Futurabond U (FBU) and Adhese Universal (ADU) – applied by both ER and SE modes, showed similar and acceptable clinical performance. Less satisfying marginal adaptation (p<0.05) were registered when applied the 1SA (Futurabond DC; FBDC, control group), by both SE and SE-EE modes, than the UAs. The overall success rates (p>0.05) were: 93.9% (FBDC-SE), 97.0% (FBDC-SE-EE,  FBU-ER, FBU-SE) and 100.0% (ADU-ER, ADU-SE). Five (2.5%) restorations (2 with FBDC-SE=2; and one with each FBDC-SE-EE, FBU-ER and FBU-SE) were missing due to retention lost (p>0.05). The success and retention rates were similar and not dependent on materials or adhesion modes. The UAs revelled better clinical outcomes than the 1SA (FBDC), particularly in SE mode.”

Line 68: since the authors refer to the use of CONSORT guidelines in study planning and reporting, the CONSORT checklist should be added as a supplementary file.

Answer: Authors added the table of this RCT CONSORT guidelines 2010, as supplementary file as indicated by reviewer.

Section 2.2, lines 79-84: please briefly describe how the parameters were evaluated, for instance, good/bad oral hygiene, high risk of caries or pulp injuries… I saw the one-year publication, and it’s not described there.

Answer: We applied the simplified oral hygiene index (OHI-s; Greene & Vermion, 1964). We considered subjects with very good and good oral hygiene those who presented OHI-S of 0.0-0.2 and of 1.3–3.0. Subjects who do presented OHI-S of 3.1-6.0 were considered as having poor oral hygiene, and were not to be included in this RCT, according to the exclusion criteria described in section 2.2.  During the recruitment and assessment  procedures, we do not had any subject classified as poor oral hygiene (as described in excluded subjects, in Figure 1).

The oral hygiene parameter was not surprising because subjects with NCCLs usually had adequate behaviours regarding prevention and control of caries, being the loss of cervical hard tissues mainly due to mechanical or chemical disturbs.

We used the International Detection and Assessment System II (ICDAS II) to assess “high risk of caries or pulp injuries”. We considered that subjects, with at least one clinical (oral examination) risk factor, that was, subjects with recent caries experience and presence of active caries lesion(s), as high risk of caries or pulp injuries. Those, were not to be included in this RCT, according to the exclusion criteria described in section 2.2.  During the recruitment and assessment  procedures, we do not had any subject classified as high risk of caries or pulp injuries (as described in excluded subjects, in Figure 1). 

We re-wrote the text, to briefly describe those parameters, and added the citations in section 2.2, and references as follows:

In our RCT we also considered as exclusion criteria subjects with poor oral hygiene or with high risk of caries or pulp injuries. The simplified oral hygiene index (OHI-s) was applied during recruitment, to include only subjects with very good or good oral hygiene. We considered subjects with “very good” and “good” oral hygiene those who presented OHI-S of 0.0-0.2 and of 1.3–3.0, respectively. Subjects who do presented OHI-S of 3.1-6.0 were considered as having poor oral hygiene [30] .

We used the International Detection and Assessment System II (ICDAS II) to assess “high risk of caries or pulp injuries”. We considered that subjects with at least one clinical risk factor, that was, subjects with recent caries experience (ICDAS II, scored different of 00) and presence of active caries lesion(s), as high risk of caries or pulp injuries. Subjects with low risk of caries, were those who do not had none clinical caries (ICDAS II, scored 00) risk factors [31]

During the recruitment, assessment and eligibility procedures, we do not found any subject classified as poor oral hygiene or high risk of caries or pulp injuries so, we do not had to exclude subjects due to those clinical conditions.

Line 84: the number of included subjects is a result and should be moved to the appropriate section.

Answer: As performed as indicated by reviewer 1. We deleted the sentence from the section 2.2. In the first paragraph of results (section 3) is described the 35 participants included.   

Section 2.3: were the cavities prepared in any way? Was sclerotic dentin partially removed to improve the adhesion?

Answer: We re-wrote the text in section 2.3 (now, section 2.5 of the reviewed manuscript) about cavities preparations and procedures performed:

No cavity preparation/design such as, enamel bevel, enamel/dentin roughened with diamond burns or mechanical retentions were performed in NCCLs. A conservative dental hard tissues concept and the NCCLs design was considered as acceptable clinical conditions to achieve, possibly, the gold of adhesives restorations. After shade selection, all NCCLs cavities and were properly cleaned with pumice and water in a rubber cup followed by rinsing and drying.”

Section 2.4: please describe how/what was accessed to attribute the scores of excellent, good, sufficient… in each category (esthetic, functional and biological). This part is unclear. For instance, the authors refer that the staining of the margins was evaluated for the esthetic property. If staining was present, which score was attributed? Or it was dependent on the staining intensity?

Answer: Authors followed the recommendations for conducting controlled clinical studies of dental restorative materials, using the FDI criteria [28]. We added a general description on how/what was accessed to attribute the scores to each criteria/category. We did not describe or add in the text all scores description for each category, because all of them are fully described according to the appropriate citation [28] regarding the FDI criteria.

For example, for staining margin if scored 1 (EX) it means that no marginal staining was detected. For stanning margin parameter, minor discoloration were only visible during inspection with a mirror and operating light (Score 2, GO), while severe discoloration were visible at a speaking distance of 60–100 cm (score 4, UNS).  Restoration fracture and retention parameters were directly assessed a quantified chart was performed to characterize the localization of cracks or chipping (fracture with loss of material at the surface of the restoration). All restoration that were present with no fractures, cracks or Chipping scored 1 (EX); All restoration that were missing or with bulk fracture with probable gap >250 μm with or without partial loss of the restoration, scored 5 (PO). 

If reviewer consider that more information is needed, that is fully described in the reference cited, we could try to add a supplementary file with the table of FDI criteria, with each score description, for each parameter.

We reviewed and added the text, as indicated by the reviewer, to clarify the ranking scores and categories of clinical acceptance:  

“As recommendations for conducting controlled clinical trials [28], each examiner(s) used a two-step approach for assigning scores for each parameter. First, each restoration was assessed to determine the level of clinical acceptability (Score 1, 2 or 3) or unacceptability (Score 4 and 5), for each parameter in each of the categories (aesthetic, functional, biological). Secondly, a further distinction was made, between an excellent (EX, score 1), Good (GO, score 2) and Sufficient (SS, score 3) for each aesthetic (staining margin), functional (fractures/retention and marginal adaptation) and biological (postoperative (hyper-) sensitivity and caries recurrence) property. The restorations clinically unacceptable were scored as 4, Unsatisfactory (UNS; needed repair for prophylactic reasons) or as 5, Poor (PO; needed replacement).

The score 1 (EX) indicated that the quality of the restoration was excellent/fulfilled all quality criteria, and the tooth and/or surrounding tissues were adequately protected. The score 2 (GO) was registered when the quality of the restoration was still highly acceptable, though one or more criteria deviated from the ideal. The restoration could be modified by polishing and upgraded to an ‘excellent’ rating but this was not normally necessary. There was no risk of damage to the tooth and/or the surrounding tissues. The Score 3 (SS) indicated that the quality of the restoration was sufficiently acceptable but with minor shortcomings. Because of their location/extent, however these could not be eliminated without damage to the tooth, though no adverse effects were anticipated. The restorations that scored 4 were UNS, but repaired whereas, those that scored 5 (PO) were replaced [28].”

-Figure 1: I suggest this figure be moved to the results section since it presents results from the 2-years follow-up, including the dropouts and the restorations clinical evaluation.

Answer: As suggested by reviewer 1, the Figure 1 was moved to the results section.

Section 2.5: if no data existed to perform a sample calculation, did the authors consider doing first a pilot study to acquire the necessary data to calculate the sample?

Answer: No, that was not considered as it would take at least two years to get some results and for that we would need the same time of bureaucratic work as for the overall clinical trial study. Therefore, we decided to do the clinical trial on the basis of the (almost inexistent) information we had, at that time. Nevertheless, the assumptions considered for the sample size calculation allowed for detection of significant differences in the time frame considered (two-years), which means they were not unfounded.

-I suggest the materials and methods section be reorganized. For instance, the sample size determination is described after the patient’s enrolment and follow-up examinations. I recommend an organization corresponding to the study prosecution be adopted.

Answer: As suggested materials and methods sections were reorganized as follows:

2.1. Trial Design

2.2. Participants

2.3. Sample Size

2.4. Randomisation and Blinding

2.5. Interventions

2.6. Follow-up Examinations and Outcomes

2.7. Statistical Analysis

Table 2: what is the meaning of Me? Median? Please add it to the table caption.

Answer: The term Me, means Median and was added to the table caption.

Section 3.1 and table 3: the authors refer to one-year follow-up results that have been already published. The results from the one-year evaluation are not results from the present study and should not be included in the results section. Besides, no reference is made to the previous article, so this can be seen as plagiarism. You can refer to them in the introduction or discussion sections with the appropriate citation.

Answer: Authors refer to the baseline, one-year recall restorations/scores results in table 3 with the purpose to describe the restorations distribution and clinical scores in the time frame, that was, during the 2-years follow-up. But we do agree with the reviewer, that are not the key-results for the 2-years follow-up, and this manuscript section 3.1, although they belong to the time frame of the two years results.

We performed as reviewed suggested, as follow:

  • We added in table 3 the appropriate citation [23] of the 1st year results already published.
  • We re-wrote the results in section 3.1, only with the key-results of the 2-years follow-up:

Fractures and retention clinical acceptance decreased in FBU-SE (P=0.041) at the 2-year recall. Marginal adaptation decreased in the FBDC-SE (P=0.009), in FBU (ER; P=0.038 and SE; P=0.024) and in ADU (ER; P=0.038 and SE; P=0.023) at  the 2-year recall.”

  • We added the following sentence in discussion, section 4, and the citation of the previous article:

Fractures and retention clinical acceptance significantly decreased in FBDC-SE at the 1-year recall [23] and in FBU-SE at the 2-year recall. Marginal adaptation decreased in FBDC-SE at the 1-year [23] and the 2-year recalls, but with the UAs (with both, ER and SE mode), the FBU and the ADU this occurrence were significantly registered, only at the 2-year recall.

Table 4 and section 3.5: how was the age of 56 determined as the cutoff to comparisons?

Answer: As showed on Table 2 and described in the first paragraph of discussion, section 4, our RCT involved “thirty-eight participants, 21 male and 17 female, with a median age of 56 years (range 24-63 years)”.

Considering the dimension of the population involved, the median age of participants gives a slightly better picture of what the age distribution itself looks like: when we see a median of 56 years, for example, we know that half the population with the NCCLs adhesives restorations, in this RCT, is older than 56 and we can infer some things about the influence of the adhesive´s restorations performance and the participants/dental hard tissues age. For that purpose we defined the median age of participants as the cut-off for statistical analysis and comparisons.

We added the following sentence in section 3.5:

The median age of the participants was defined as the cut-off for the comparisons.”

Lines 315-317: the authors stated that “the restorations that scored as clinically unsatisfactory at the first-year recall were no longer included in the study”. This sentence is unclear. Does this mean they were excluded from the analysis and not accounted for? Or that they were not clinically replaced?

Answer: The restorations that scored as clinically unsatisfactory (UNS) at the first-year recall were not excluded from the RCT, because they were counted as cumulative failure at the 2-years analysis.

According to the human and good clinical practices guidelines for clinical trials, all restorations that are/were considered with relative (UNS) or absolute failures (PO) were clinically repaired or replaced. As described in discussion, section 4, “ At one-year recall, one restoration with FBDC-SE and other with ADU-ER scored as unsatisfactory for fractures and retention (Table 3) and were repaired.”

We re-wrote the text, to clarity the sentence as follow:

At one-year recall, one restoration with FBDC-SE and other with ADU-ER scored as unsatisfactory for fractures and retention (Table 3) and were clinically repaired. In our study, all the restorations that scored as clinically unsatisfactory for FDI criteria at the first year recall were accounted, as cumulative failure, at the 2-years follow-up analysis."

Reviewer 2 Report

This research may have significant relevance in the conservative dentistry field, considering: the type of study conducted, the relevance of the data and the large incidence of non-carious cervical lesions.

just a few considerations I want to ask:

  • Is Possible representing failures in groups through Kaplan Meier survival curves? Is it possible to calculate the Hazard Ratio between the different groups?
  • Why did you not take into account the USPHS criteria for evaluating restorations?
  • I would add in the introduction a small paragraph \ chapter on the nature / etiology of non-carious lesions and on the problems of adhesion of the materials related to them.

Author Response

Response to Reviewer 2 Comments

Just a few considerations I want to ask:

Is Possible representing failures in groups through Kaplan Meier survival curves? Is it possible to calculate the Hazard Ratio between the different groups?

Answer: As explained in the section 2.7, lines 173-177 of the initial submitted manuscript “We conducted pairwise comparison (performance, success, and retention rates) from baseline up to the 2-year follow-up between research groups, SE and SE-EE/ER adhesion modes, and adhesives with Kaplan-Meier survival curves and log-rank tests.” We didn´t present the calculations for each of the 6 different groups as the hypothesis under test didn´t ask for that. Information regarding p-values results (but not the hazard ratios) is presented in the Table 4.

Why did you not take into account the USPHS criteria for evaluating restorations?

Answer: Yes, we took into account the USPHS criteria. In this RCT (Clinical trial.gov) register and design, we do applied both criteria (FDI and USPHS) for restorations examination. However, we do considered that minor effects observed in NCCLs restorations are more specific and detailed by FDI criteria particularly, in short-mean terms, as the one- and the 2-years follow-ups.

More recently, in a scoping review about the use of FDI criteria in clinical trials on direct dental restorations (Marquillier and Collegues, 2018) the authors concluded that FDI criteria were reported as practical (various and freely selectable), relevant (sensitive as well as appropriate to current restorative materials and clinical studies design), standardized (making comparisons between investigations easier). Investigators should go on using them for a better standardization of their clinical judgment, allowing comparisons with other studies.

In this RCT we followed the the recommendations for conducting controlled clinical studies of dental restorative materials, using the FDI criteria [28]. Although examiners have done, independently , both examinations, by USPHS and by FDI criteria, the scores rating (1, 2, 3,4 and 5) of FDI criteria also could be corresponded to Ryge´s rating, in some way, and according to the citation [28]. Alfa rating corresponded to FDI score 1 (EX) and 2 (GO); Bravo rating to FDI score 3 (SS). Scores 4 and 5 corresponded to Ryge´s Charlie and Delta scoring.

We took in account and appreciate this comment done by the reviewer. In future follow-ups (more than 2-years reports) we will try to perform the publications describing both evaluation criteria results.

  • I would add in the introduction a small paragraph \ chapter on the nature / etiology of non-carious lesions and on the problems of adhesion of the materials related to them.

Answer: Authors added in introduction a small paragraph on etiology and main adhesion  problems related to NCCLs restorations, as suggested by reviewer:

“Non-carious cervical lesions (NCCLs) involves a cervical teeth wear, with loss of hard tissues unrelated to carious disease. Commonly, lesions are generated by multifactorial conditions related with changes in lifestyle and diet. Aging is also related with a higher teeth cervical wear. The clinical appearance of NCCLs can vary depending on the type and severity of the etiologic factors involved. Adhesive restoration and suitable control of etiological factors are appropriate clinical procedures to prevent the progression of those cavity lesions [11]. Composite restoration of NCCLs are challenger due to the adhesive reactions with enamel and various dentin tissues, the biomechanical aspects of the cervical area and, the limitations on the access and isolation of the operative field [12]. The impregnation of the dentine substrate by resin monomers and the stability of the bonded interface are of paramount importance to clinical performance [13].”

Round 2

Reviewer 1 Report

The authors present the 2-year results of a randomized clinical trial, reporting the performance of two universal adhesives applied in self-etch or etch-and-rinse modalities. The study protocol is described, along with the restorations clinical results and patient´s characteristics. Comparisons between the study groups are made and regarding the participant´s and the restored lesions characteristics.

The performed modifications improved the manuscript quality. However, I still have a minor concern.

I suggest the sample size (2.3) be described before the participants (2.2).

Table 3: the authors refer to one-year follow-up results that have been already published. Do you have permission from the publisher to reproduce the one-year results? Again, I suggest this information be removed since the present manuscript aims to report the two-year results, and a description of the main findings of the one-year recall is made in the introduction section.

Author Response

I suggest the sample size (2.3) be described before the participants (2.2).

Answer: We performed as suggested and moved sample size (2.2) before the participants (2.3).

Table 3: the authors refer to one-year follow-up results that have been already published. Do you have permission from the publisher to reproduce the one-year results? Again, I suggest this information be removed since the present manuscript aims to report the two-year results, and a description of the main findings of the one-year recall is made in the introduction section.

Answer: We described in table 3, the data of both from baseline and first-year next to the second year follow-up in order to clarify the progression of the evaluations of the NCCL restorations. We also add the reference of the previous paper in the table 3 title.  Those that want to use data results from our first-year follow-up have to mention the previous paper, publishing in Open Access. So, that can be accessed by the appropriate audience. As commented by reviewer, as the 1-year main findings are made in introduction section we do removed those data from table 3, as suggested. We do acknowledge this concern and also agree.

Reviewer 2 Report

The authors answered all the questions asked. I believe the manuscript can be published in this form.

Author Response

Thank you in advance, 

Kind regards

Round 3

Reviewer 1 Report

The authors have addressed all my comments. Good work.